


**Simulating Lightning NO$_X$ Production in CMAQv5.2 Using mNLDN, hNLDN,**
**and pNLDN Schemes: Performance Evaluations**
Daiwen Kang[1]*, Kristen Foley[1], Rohit Mathur[1], Shawn Roselle[1], Kenneth Pickering[2], and Dale
Allen[2]
[1]Computational Exposure Division, National Exposure Research Laboratory, U.S.
Environmental Protection Agency, Research Triangle Park, NC 27711, USA
[2]Department of Atmospheric and Oceanic Science, University of Maryland, College Park, MD,
USA
*Corresponding author: Daiwen Kang, US EPA, 109 T.W. Alexander Drive, Research Triangle Park, NC
27711, USA. Tel.: 919-541-4587; fax: 919-541-1379; e-mail: kang.daiwen@epa.gov





# Abstract

This study assesses the impact of the lightning $NO_X$ ($LNO_X$) production schemes in the
CMAQ model (Kang et al., 2019) on ground-level air quality as well as aloft atmospheric
chemistry through detailed evaluation of model predictions of nitrogen oxides ($NO_x$) and ozone
($O_3$) with corresponding observations for the U.S. For ground-level evaluations, hourly $O_3$ and
$NO_x$ from the US EPA's AQS monitoring network are used to assess the impact of different
LNOx schemes on model prediction of these species in time and space. Vertical evaluations are
performed using ozonesonde and P-3B aircraft measurements during the DISCOVER-AQ
campaign conducted in the Baltimore/Washington region during July 2011. The impact on wet
deposition of nitrate is assessed using measurements from the National Atmospheric Deposition
Program's National Trends Network (NADP/NTN). Compared with the base model (without
LNOx), the impact of LNOx on surface $O_3$ varies from region to region depending on the base
model conditions. Overall statistics suggest that for regions where surface $O_3$ mixing ratios are
already overestimated, the incorporation of additional $NO_x$ from lightning generally increased
model overestimation of mean daily maximum 8-hr (DM8HR) $O_3$ by 1-2 ppb. In regions where
surface $O_3$ is underestimated by the base model, LNOx can significantly reduce the
underestimation and bring model predictions close to observations. Analysis of vertical profiles
reveals that $LNO_x$ can significantly improve the vertical structure of modeled $O_3$ distributions by
reducing underestimation aloft, and to a lesser degree decreasing overestimation near the surface.
Since the base model underestimates the wet deposition of nitrate in most regions across the
modeling domain except the Pacific Coast, the inclusion of LNOx leads to reduction in biases
and errors and an increase in correlation coefficients at almost all the NADP/NTN sites. Among
the three LNOx schemes described in Kang et al. (2019), the hNLDN scheme, which is
implemented using hourly observed lightning flash data from National Lightning Detection
Network (NLDN), performs best for the ground-level, vertical profiles, and wet deposition
comparisons except that for the accumulated wet deposition of nitrate, the mNLDN scheme (the
monthly NLDN-based scheme) performed slightly better. However, when observed lightning
flash data are not available, the linear regression-based parameterization scheme, pNLDN,
provides an improved estimate for LNOx compared to the base simulation that does not include
LNOx.



## 1. Introduction

The potential importance of $NO_X$ produced by lightning ($LNO_X$) on regional air quality was recognized more than two decades ago (e.g. Novak and Pierce, 1993), but $LNO_X$ emissions have only been added to regional chemistry and transport models during the last decade (e.g. Allen et al., 2012; Kaynak et al., 2008; Koshak et al., 2014; Smith and Mueller, 2010; Koo et al., 2010) owing in part to the limited understanding of this $NO_X$ source (Schumann and Huntrieser, 2007; Murray, 2016; Pickering et al, 2016). As a result of efforts to reduce anthropogenic NOx emissions in recent decades (Simon et al., 2015; https://gispub.epa.gov/air/trendsreport/2018), it is expected that the relative contribution of $LNO_X$ to the tropospheric $NO_X$ burden and its subsequent impacts on atmospheric chemistry will increase in the United States and other developed countries (Kang and Pickering, 2018). The significant impact of $LNO_X$ on surface air quality was earlier reported by Napelenok et al. (2008), in that low-biases in upper tropospheric NOx in Community Multiscale Air Quality Model (CMAQ) (Byun and Schere, 2006) simulations without $LNO_X$ emissions made it difficult to constrain ground-level NOx emissions using inverse methods and Scanning Imaging Absorption Spectrometer for Atmospheric Cartography (SCIAMACHY) $NO_2$ retrievals (Bovensmann et al., 1999; Sioris et al., 2004; Richter et al., 2005). Appel et al. (2011) and Allen et al. (2012) reported that $NO_3^-$ wet deposition at National Atmospheric Deposition Program (NADP) sites was underestimated by a factor of two when LNOx was not included.

$LNO_X$ production and distribution were parameterized initially in global models (e.g. Stockwell et al., 1999; Labrador et al., 2005) relying on the work of Price and Rind (1992) and Price et al. (1997) in that lightning flash frequency was parameterized as a function of the maximum cloud-top-height. Other approaches for $LNO_X$ parameterization include a combination of latent heat release and cloud-top-height (Flatoy and Hov, 1997), convective precipitation rate (e.g. Allen and Pickering, 2002), convective available potential energy (Choi et al., 2005), or convectively induced updraft velocity (Allen et al., 2000; Allen and Pickering, 2002). More recently, Finney et al. (2014, 2016) adopted a lightning parameterization using upward cloud ice flux at 440hPa (based upon definitions of deep convective clouds in the International Satellite Cloud Climatology Project (Rossow et al., 1996)) and implemented it in the United Kingdom Chemistry and Aerosol model (UKCA). With the availability of lightning flash data from the



National Lightning Detection Network (NLDN) (Orville et al., 2002), recent $LNO_X$
parameterization schemes started to include the observed lightning flash information to constrain
$LNO_X$ in regional Chemical Transport Models (CTMs) (Allen et al., 2012). In Kang et al. (2019),
we described the existing $LNO_X$ parameterization scheme that is based on the monthly NLDN
(mNLND) lightning flash data, and an updated scheme using hourly NLDN (hNLDN) lightning
flash data in the CMAQ lightning module.  In addition, we also developed a scheme based on
linear and log-linear regression parameters using multiyear NLDN observed lightning flashes
and model predicted convective precipitation rate (pNLDN). The preliminary assessment of
these schemes based on total column $LNO_X$ suggests that all the schemes provide reasonable
$LNO_X$ estimates in time and space, but during summer months, the mNLDN scheme tends to
produce the most and the pNLDN scheme the least $LNO_X$.

The first study on the impact of $LNO_X$ on surface air quality using CMAQ was conducted

by Allen et al. (2012) and followed by Wang et al. (2013) with different ways for parameterizing
$LNO_X$ production and different model configurations. In this study, we present performance
evaluations using each of the $LNO_X$ production schemes (mNLDN, hNLDN, pNLDN) described
by Kang et al. (2019) to provide estimates of $LNO_X$ in CMAQ. In addition to examination of
differences in air quality estimates between these schemes, we compare the model predictions to
base model estimates without LNOx and evaluate the estimates from all of the simulations
against surface and airborne observations.

Section 2 describes the model configuration, simulation scenarios, analysis methodology,

and observational data.  Section 3 presents the analysis results and Section 4 presents the
conclusions.

## 108    2.  Methodology

### 109    2.1 The CMAQ model and simulation configurations

The three $LNO_x$ production schemes described in Kang et al (2019) were incorporated

into CMAQ v5.2 (Appel et al. 2017; doi:10.5281/zenodo.1167892). The chemical mechanism
used was CB6 (Yarwood et al., 2010) and the aerosol module was AERO6 (Nolte et al., 2015).



The meteorological inputs were provided by the Weather Research and Forecasting (WRF)
model version 3.8 and the model-ready meteorological input files were created using version 4.2
of the meteorology–chemistry interface processor (MCIP; Otte and Pleim, 2010).
The modeling domain covers the entire contiguous United States (CONUS) and
surrounding portions of northern Mexico and southern Canada, as well as the eastern Pacific and
western Atlantic oceans. The model domain consists of 299 north–south grid cells by 459 east–
west grid cells utilizing 12 km x 12 km horizontal grid spacing, 35 vertical layers with varying
thickness extending from the surface to 50 hPa and an approximately 10m midpoint for the
lowest (surface) model layer. The simulation time period covers the months from April to
September 2011 with a 10-day spin-up period in March.
Emission input data were based on the 2011 National Emissions Inventory
(https://www.epa.gov/air-emissions-inventories). The raw emission files were processed using
version 3.6.5 of the Sparse Matrix Operator Kernel Emissions (SMOKE;
https://www.cmascenter.org/smoke/) processor to create gridded speciated hourly model-ready
input emission fields for input to CMAQ. Electric generating unit (EGU) emissions were
obtained using data from EGUs equipped with a continuous emission monitoring system
(CEMS). Plume rise for point and fire sources were calculated in-line for all simulations (Foley
et al., 2010). Biogenic emissions were generated in-line in CMAQ using BEIS versions 3.61
(Bash et al., 2016). All the simulations employed the bidirectional (bi-di) ammonia flux option
for estimating the air–surface exchange of ammonia.
There are four CMAQ simulation scenarios for this study: 1) simulation without $LNO_X$
(Base), 2) simulation with $LNO_X$ generated by the scheme based on monthly information from
the NLDN (mNLDN), 3) simulation with $LNO_X$ generated by scheme based on hourly
information from the NLDN (hNLDN), and 4) simulation with $LNO_X$ generated by the scheme
parameterizing lightning emissions based on modeled convective activity (pNLDN) as described
in detail in Kang et al. (2019). All other model inputs, parameters and settings were the same
across the four simulations. The vertical distribution algorithm is the same for all the $LNO_X$
schemes as also described in Kang et al. (2019).





## 2.2 Observations and analysis techniques


To assess the impact of LNOx on ground-level air quality, output from the various CMAQ
simulations were paired in space and time with observed data from the EPA's Air Quality
System (AQS; https://www.epa.gov/aqs) for hourly $O_3$ and $NO_X$. To evaluate the vertical
distribution, measurements of trace species from the Deriving Information on Surface Conditions
from Column and Vertically Resolved Observations Relevant to Air Quality (DISCOVER-AQ;
http://www.nasa.gov/missions/discover-aq) campaign conducted in the Baltimore/Washington
region (e.g., Crawford and Pickering, 2014; Anderson et al., 2014; Follette-Cook et al., 2015)
were used. During this campaign, the NASA P-3B aircraft measured trace gases including $O_3$,
NO, and $NO_2$. Vertical profiles were obtained over seven locations – Beltsville (Be), Padonia
(Pa), Fairhill (Fa), Aldino (Al), Edgewood (Ed), Essex (Es), and Chesapeake Bay (Cb) from
approximately 0.3 to 5 km above ground level during P-3B flights over 14 days in July 2011.
During this same period, ozonesonde measurements were taken that extended from ground level
through the entire model column at two locations (Beltsville, MD and Edgewood, MD shown in
Figure 1). Inclusion of LNOx estimates in the CTM simulations also has an important impact on
model estimated wet deposition of nitrate.  Therefore, assessment was also performed using data
from the National Atmospheric Deposition Program's National Trends Network (NADP/NTN,
http://ndp.slh.wisc.edu/ntn).
Since lightning activity as well as LNOx exhibit distinct spatial variations (Kang and
Pickering, 2018), analysis was conducted for the model domain over the contiguous United
States, and then for each region as shown in Figure 1. Emphasis is placed on two regions,
Southeast (SE) and Rocky Mountains (RM), where lightning activity is more prevalent and
LNOx has the greatest impact on model predictions as shown in Results - increasing model bias
in the SE and decreasing bias in the RM. The commonly used statistical metrics, Root Mean
Square Error (RMSE), Normalized Mean Error (NME), Mean Bias (MB), Normalized Mean
Bias (NMB), and Correlation Coefficient (R), in the model evaluation field as defined in Kang et
al. (2005) and Eder et al. (2006) were calculated to assess the basic performance differences
among all the model cases for their ground-level air quality predictions.





## 3.  Results
### 3.1 Ground-level evaluation for O$_3$ and NO$_X$
### 3.1.1 Statistical performance metrics
Tables 1 and 2 display the statistical model performance metrics for daily maximum 8-hr
(DM8HR) O$_3$ and daily mean NOx mixing ratios over the domain and each analysis region for
all four model cases in July 2011 (Base, mNLDN, hNLDN, and pNLDN). The best performance
metrics among the model cases are highlighted in bold. As shown in Table 1, for DM8HR O$_3$,
the Base simulation has the lowest MB and NMB values over the Domain, while hNLDN
produced the smallest RMSE and NME values. mNLDN generated the largest values for both
error (RMSE and NME) and biases (MB and NMB), followed by pNLDN. More importantly, all
model cases with LNOx exhibit slightly higher correlation coefficients than the Base simulation,
suggesting the importance of including the contributions of this source for improving the spatial
and temporal variability in model predictions. Additionally, the hNLDN simulation exhibited
higher correlation and lower bias and error relative to the measurements indicating the value of
higher temporal resolution lightning activity for representing the associated NO$_x$ emissions and
their impacts on tropospheric chemistry.
Examining the regional results for DM8HR O$_3$ in Table 1, the statistical measures indicate
that in the Northeast (NE), hNLDN outperformed all other model cases with the lowest errors
and biases and highest correlation coefficient. In Southeast (SE), the Base performed better with
the lowest errors and mean biases, but the correlation coefficient (R) value for hNLDN is slightly
higher. Among all the LNOx cases, mNLDN produced the worst statistics in this region.
Historically, CTMs tend to significantly overestimate surface O$_3$ in the Southeast US (Lin et al.,
2008l Fiore et al., 2009l Brown-Steiner et al., 2015; Canty et al., 2015), and this is speculated to
be driven in part by an overestimation of anthropogenic NOx emission estimates. Thus, even
though lightning is known to impact ambient air quality, including this additional NOx source
can worsen model performance in some locations and time periods due to other errors in the
modeling system.  As noted in Table 1, for SE, the MB values increased by about 1.6 ppb in
mNLDN and less than 1 ppb in hNLDN and pNLDN. Nevertheless, the correlation coefficients
for mNLDN and pNLDN are almost the same with the Base, and hNLDN was slightly higher

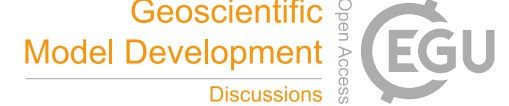



(0.77 compared to 0.76). These correlations indicate that even though additional NOx increases
the mean bias, when it is added correctly in time and space, as with the case of hNLDN, the
spatial and temporal correlation are improved. In Upper Midwest (UM), the lowest errors and
biases among the model cases are associated with hNLDN, while the worst performance is with
mNLDN. In the Lower Midwest (LM), hNLDN performed comparable with the Base, with
hNLDN having the highest correlation and lowest mean errors, while the Base has the lowest
mean biases. Rocky Mountain (RM) is the only region that shows an underestimation of
DM8HR $O_3$. In this region all the model cases with LNOx outperformed the Base case in all the
metrics. Among the three model cases with LNOx, mNLDN produced the lowest MB and NMB
values, while hNLDN had the lowest RMSE and NME, and the highest correlation. In the Pacific
Coast (PC) region, lightning activity is generally very low compared to other regions (Kang and
Pickering, 2018). All model cases with LNOx outperformed the Base case, especially hNLDN
which had the lowest mean error and bias and highest correlation among all the cases.
Most of the $NO_x$ produced by lightning is distributed in the middle and upper troposphere
with only a small portion being distributed close to the surface.  As a result, the impact on
ground-level $NO_x$ mixing ratios is small. Table 2 shows all the model cases produced similar
statistics for the daily mean $NO_x$ mixing ratios at AQS sites across the domain and within all the
subregions. Although the changes in model performance are small, the model cases with LNOx
exhibit similar or slightly better performance than the Base case.

### 223   3.1.2 Time series

Figure 2 presents the timeseries of regional-mean observed and modeled DM8HR $O_3$ for
the entire domain and the SE and RM regions during July 2011. Over the domain and in SE, all
the model cases overestimate the mean DM8HR $O_3$ mixing ratios on all days with the Base being
the closest to the observations. hNLDN is almost the same as the Base with slightly higher
values on some days. Among all the cases, mNLDN produced the highest values on almost all
days through the month, on the order of 1-2 ppb higher than the Base. In contrast, in the RM
region, the Base significantly underestimates DM8HR $O_3$ mixing rations on all the days during
the month, while all model cases with LNOx improved model predictions relative to
observations in the region. Among the three model cases with LNOx, mNLDN produced the
lowest bias for all the days, closely followed by hNLDN.





Figure 3 displays the average daily mean NOx mixing ratios at AQS sites over the same
regions as in Figure 2. On most of the days in July 2011, over the domain and in SE, the model
cases overestimate NOx values, and on almost half of the days, the overestimation is significant
(up to 100%). As noted in Table 2, on average, the overestimation is ~17% over the domain and
~43% in SE. However, in RM, the predicted NOx mixing ratios closely follow the daily
observations and on average the modeled and observed magnitude is almost identical (~3%
difference). All the model cases, with or without LNOx, produced almost the same mean NOx
mixing ratios at the surface. However, the different cases produce different levels of LNOx in the
middle and upper troposphere, resulting in differences in $O_3$ production and transport which
impact ground-level $O_3$ levels. We further explore these features in Section 3.2 which presents
evaluation of modeled vertical pollutant distributions.

### 3.1.3 Diurnal variations

Diurnal plots are used to further examine differences in model evaluation for $O_3$ and
$NO_x$. Figure 4 shows the mean diurnal profiles for hourly $O_3$ and $NO_x$ over the entire domain,
SE, and RM. On a domain mean basis, all model cases overestimate $O_3$ during the daytime
hours, while in the SE, the overestimation spans all the hours. In RM, the model cases
significantly underestimate $O_3$ across all the hours except for a few early morning hours, when
the model predicted values are very close to the observations. Among all the model cases, as
expected, the most prominent differences occurred during the midday hours when the
photochemistry is most active. However, the difference between hNLDN (and mNLDN) and the
Base is also significant during the night in the RM region, even though the $O_3$ levels are low.
This may be attributed to NOx-related nighttime chemistry in part caused by freshly released NO
by cloud-to-ground lightning flashes. The diurnal variations of NOx are similar over the domain
and in the regions for all model cases. Appel et al. (2017) reported a significant overestimation of
NOx mixing ratios at AQS sites during nighttime hours and underestimation during daytime
hours.  The bias pattern is identical for all of the LNOx model cases evaluated here (Figure 4).

### 3.1.4 Spatial variations

Figure 5 shows the impact of the different $LNO_x$ schemes on model performance for
DM8HR $O_3$ at AQS sites. The spatial maps show the difference in absolute MB between the

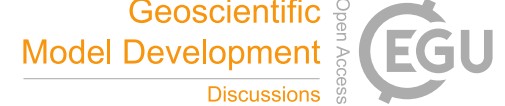



cases with lightning NO$_x$ emissions and the Base and is calculated as follows. First, the absolute
MB was calculated at each site for each case, e.g. $|MB_{[Base - Obs]}|$, then the difference in absolute
MB was calculated between model cases, e.g. $|MB_{[hNLDN-Obs]}|$ - $|MB_{[Base - Obs]}|$. The histograms of
the differences in absolute MB between model cases in Figure 5 are provided to show the
distribution of the change in model performance across space, i.e. the frequency of an
improvement in model performance versus a degradation in model performance between cases.
As shown in Figure 5, the mNLDN shows increased model bias in the east US and along the
California coast, but reduced model bias in the RM. At a majority of the AQS sites, it increases
the model bias (only decreases at 26.8% (346) of the sites). The hNLDN also significantly
reduces model bias in the RM with a moderate increase in the SE. Overall, in the hNLDN, the
mean bias decreased at 61.2% (791) of AQS sites. Similar to mNLDN, increases in mean bias
are noted at 29.3% (378) of the AQS sites in the pNLDN simulation. As noted in the histograms,
the distribution of the model bias in the pNLDN is much narrower than both mNLDN and
hNLDN, eliminating the large bias increases in mNLDN and the significant bias decreases in
hNLDN.

### 3.2 Vertical evaluation for O$_3$ and NO$_X$


#### 3.2.1    Ozone-sonde observations


A large source of uncertainty in the specification of LNOx is its vertical allocation, which
can impact the model's ability to accurately represent the variability in both chemistry and
transport. To further assess the impact of the vertical LNOx specification on model results, we
compared vertical profiles of simulated model O$_3$ with extensive ozonesonde measurements
available during the study period. Figure 6 presents the vertical profiles for O$_3$ sonde
measurements and paired model estimates of all model cases at Beltsville, MD and Edgewood,
MD. At each location, observations from multiple days are available (one or two soundings per
day) during the 2011 DISCOVER-AQ campaign in July 2011. The model evaluation was limited
to days where the inclusion of LNOx has an obvious impact (the vertical profile lines can be
separated) on the model estimates (July 21, 22, 28 and 29 at Beltsville, and July 21, 22, 28, 29,
and 30 at Edgewood). We paired the observed data with model estimates in time and space and
averaged the model and observed values at each model layer. Only data below 12 km altitude are
plotted in Figure 6 to exclude possible influence of stratospheric air on O$_3$. As can be seen in



Figure 6, at both locations the Base case underestimates $O_3$ mixing ratios from around 1 km
upwards, but overestimates closer to the surface. When LNOx is included in the simulations, the
predicted $O_3$ mixing ratios increase relative to the Base case starting around 2km, with greater
divergence from the Base case at higher altitudes. The two model cases, hNLDN and mNLDN,
produced similar $O_3$ levels until about 6 km, but above that altitude the mNLDN ozone mixing
ratios were higher. All the model cases with LNOx performed much better aloft than the Base
case. Near the surface, all the model cases overestimated $O_3$, however hNLDN had smaller bias
than the other simulations. This may be attributed to the fact that only hNLDN used the observed
lightning flash data directly, and as a result, LNOx was estimated more accurately in time and
space. This improvement in model bias at the surface is further investigated in the next section
using evaluation against P-3B measurements.
**3.2.2 P-3B measurement**
Extensive measurements of lower tropospheric chemical composition distributions over
the Northeastern U.S. are available from instruments onboard the P-3B aircraft on 14 days of the
DISCOVER-AQ campaign. We utilize measurements from one of the days (28 July 2011) with
noticeable (the mean vertical profiles of $LNO_X$ cases are separable from that of the base case)
lightning impacts, to evaluate the model simulations.  Figure 7 shows measured $O_3$ mixing ratios
overlaid on the modeled vertical time-section for 1030 – 1730 UTC. The color-filled circles
represent measured $O_3$ mixing ratios averaged over 60 seconds and the background is the model
estimated vertical profiles from the grid cells containing the P-3B flight path for that hour and
location. As indicated in the Base case (Figure 7a), the model tends to overestimate $O_3$ mixing
ratios from the surface to about 2 km, but underestimate at altitudes above 2 km. The hNLDN
reduced the overestimation below 2km, e.g. fewer grid cells with mixing ratios above 90ppb
(shown in red).  The other two cases (mNLDN, pNLDN) did not produce the same improvement
near the surface. The hNLDN also decreases the underestimation aloft compared to the Base case
with $O_3$ mixing ratios in the 55-65 ppb range (light blue colors), better matching the measured
values.  This decrease in underestimation aloft is also seen in the mNLDN case, but to a lesser
degree while the pNLDN case shows only slight improvement aloft over the Base simulation.
To further differentiate the three LNOx model cases, Figures 8-10 show the difference in
the time-sections between each of the model cases with LNOx and the Base for NO, NOx, and





$O_3$ from all the model layers along the P-3B flight path on July 28. As seen in Figure 8, the
hNLDN scheme injected most NO above 5 km and small amount near the surface, with the
maximum amount injected between 13-14 km. After release into the atmosphere, NO is quickly
converted into $NO_2$ in the presence of $O_3$, and these collectively result in the $NO_x$ ($NO+NO_2$)
vertical time-section (local production plus transport) shown in the middle panel of Figure 8.
$NO_x$ is further mixed down through the time-section and more persistent along the flight path
near the surface than is NO. As a result, significant $O_3$ is produced above 3 km and the maximum
$O_3$ difference appears between 9 and 14 km during the early afternoon hours (from 13:30 to
17:30). However, from surface to about 2 km, $O_3$ is reduced consistently across the entire period,
and this is the result of $O_3$ titration by NO from cloud-to-ground lightning flashes that must have
been transported to this layer by storm downdrafts. Since $O_3$ is significantly underestimated
above 3 km and overestimated near the surface by the Base model, the inclusion of LNOx
greatly improved the model's performance under both conditions.
Comparison of Figure 9 (mNLDN) with Figure 8 (hNLDN) reveals that the time-sections
of NO and NOx above 5 km are similar for these two cases, but they are dramatically different
near the surface. The near-surface increase in ambient NO noted in the hNLDN is absent in
mNLDN, and in fact there are some small decreases in NO, although the reason for this is
unclear.  The increase in $O_3$ aloft in the mNLDN case is similar to that seen in the hNLDN case.
However, the near-surface reduction in $O_3$ is almost absent. In the pNLDN case (Figure 10), NO
mixing ratios are much less than those in hNLDN and mNLDN in the upper layers as a result of
less column NO being generated by the linear parameterization. The resulting NOx time-section
is also smoothed. The pNLDN time-sections for NO, NOx and $O_3$ near the surface are similar to
the mNLDN case with no change or small decreases compared to the Base case. $O_3$ mixing ratios
increase by more than 30 ppb during the afternoon hours between $10 – 13$ km in the pNLDN
case, however the increase is not as intense and widespread as the other cases. In summary, the
hNLDN scheme produces estimates that are more consistent with measurements at the surface
and aloft, compared to the other simulations, reflecting the advantage of using the spatially and
temporally-resolved observed lightning flash data.  The model performance improvement for
simulated $O_3$ distributions also suggests robustness in the vertical distribution scheme when
LNOx is generated at the right time and location.





To corroborate the above time-section distributions of NO, $NO_X$, and $O_3$ in the lightning
cases, the lightning NO emissions are traced back on July 28 for each case. It is found that in all
cases, the lightning NO was injected about 200 km upwind (north-west) of the flight path. The
hNLDN case captured two injections: one occurred during the morning hours (5:00 to 7:00 am)
and the other happened during the afternoon hours (after 2:30 pm). Both mNLDN and pNLND
captured the afternoon lightning event at the later time (after 3:30 pm for mNLDN and after 4:30
pm for pNLDN) with varying intensity, but neither captured the morning lightning event, which
explains why the increase of NO and $NO_X$ in the hNLDN case (Figure 8) did not occur in the
mNLDN and pNLDN cases (Figures 9 and 10). Also note that the significant increase of NO
during the time period from 11:00 to 13:00 occurred about 5 hours after the lightning NO was
injected at about 200 km upwind in the hNLDN case.
To expand on the evaluation in Figures 7-10 which focused on measurements from July
28, 2011, we retrieved all the P-3B measurements on days with noticeable lightning impact (July
21, 22, 28, and 29). The 3-D paired observation-model data were grouped together by spiral site
and the mean biases (model – observation) were plotted in Figure 11 (a and b) for $O_3$ and NO,
respectively. The boxplots for $O_3$ in Figure 11a suggests that the Base exhibited larger bias with
greater spread (i.e. larger interquartile range) than other model cases incorporating LNOx at most
of the locations where aircraft spirals were conducted. At all locations except Aldino, the lowest
mean biases in simulated NO and $O_3$ are noted in the hNLDN simulation.

### 3.3 Deposition evaluation for nitrate

In addition to contributing to tropospheric $O_3$ formation, NOx oxidation also leads to gaseous
nitric acid and particulate-nitrate which are eventually removed from the atmosphere by dry and
wet deposition of nitrate ($NO_3^-$).  As a result, inclusion of NOx from lightning also plays an
important role in nitrogen deposition modeling. To assess the impacts of incorporating LNOx
emissions on simulated oxidized nitrogen deposition, we compared model estimated amounts of
precipitation from NTN network (http://nadp.slh.wisc.edu/ntn/) and wet deposition of $NO_3^-$ with
measurements from the NADP network (http://nadp.slh.wisc.edu/). During summer months in
2011 (June -August) the WRF model generally reproduces the observed precipitation with a
slight underestimate in the east, but the Base model simulation tends to underestimate wet



deposition of $NO_3^-$ across the domain, with the greatest underestimation in the SE and UM (See
Table 3 and Figure 12). All three LNOx simulations increase wet deposition amounts of $NO_3^-$
and decrease model bias in all regions. The bottom panel of Figure 12 shows that the mNLDN
simulation resulted in the largest increase over the base model estimates. The NMB is reduced
from -35% in the Base to -15% in mNLDN across the domain and from -32% to -2% in the SE.
The hNLDN shows very similar model performance to the mNLDN case. In contrast, the wet
deposition $NO_3^-$ estimates from the pNLDN case are only slightly higher than the Base case, and
as a result the evaluation statistics for pNLND are very similar to the Base statistics. As
discussed earlier, the mNLDN tends to produce the most LNOx among the three LNOx schemes,
thus it results in the smallest errors in terms of wet deposition of $NO_3^-$ when compared to the
Base simulation that significantly underestimated $NO_3^-$ wet deposition. It should be noted that in
addition to the LNOx contributions, errors in modeled precipitation amounts and patterns also
likely influence the underestimation of $NO_3^-$ wet deposition.

## 4. Conclusions

A detailed evaluation of lightning NOx emission estimation parameterizations available
in the CMAQ modeling system was performed through comparisons of model simulation
results with surface and aloft air quality measurements.
Our analysis indicates that incorporation of LNOx emissions enhanced $O_3$ production in
the middle and upper troposphere, where $O_3$ mixing ratios were often significantly
underestimated without the representation of LNOx. Though the impact on surface $O_3$ varies
from region to region and is also dependent on the accuracy of the NOx emissions from other
sources, the inclusion of LNOx, when it is injected at the appropriate time and location, can
improve the model estimates. In regions where the base model estimates of $O_3$ were biased
high, the inclusion of LNOx further increased the model bias; and a systematic increase is
noted in the correlation with measurements, suggesting that emissions from other sources
likely drive the overestimation. Identifying how errors in emissions inputs from different
sources interact with errors in meteorological modeling of mixing and transport, remains a
challenging but critical task. Likewise, all the LNOx schemes also enhanced the accumulated





wet deposition of $NO_3^-$, that was significantly underestimated by the base model without
LNOx throughout the modeling domain except the Pacific Coast.
Uncertainty remains in modeling the magnitude and spatial, temporal and vertical
distribution of lightning produced NOx. LNOx schemes are built on numerous assumptions
and all current schemes also depend on the skill of the upstream meteorological models in
describing convective activity. Nevertheless, these schemes reflect our best understanding
and knowledge at the time when the schemes were implemented. The use of hourly
information on lightning activity yielded LNOx emissions that generally improved model
performance for ambient $O_3$ and $NO_x$ as well as oxidized nitrogen wet deposition amounts.
As more high-quality data from both ground and satellite measurements become available,
the performance of the LNOx schemes will continue to improve.

**424     Code and data availability**

CMAQ model documentation and released versions of the source code, including all model
code used in his study, are available at https://www.epa.gov/cmaq. The data processing and
analysis scripts are available upon request. The WRF model is available for download through
the WRF website (http://www.wrf-model.org/index.php).
The raw lightning flash observation data used are not available to the public but can be
purchased through Vaisala Inc. (https:// www.vaisala.com/en/products/systems/lightning-
detection). The immediate data behind the tables and figures are available from
https://zenodo.org/record/2621096 (Kang and Foley, 2019). Additional input/output data for
CMAQ model utilized for this analysis are available upon request as well.
**Disclaimer:** The views expressed in this paper are those of the authors and do not necessarily
represent the views or policies of the U.S. EPA.

**439     Author Contribution**

**Daiwen Kang:** data collection, algorithm design, model simulation, analysis, and manuscript
writing.
**Kristen Foley**: data analysis and manuscript writing.
**Rohit Mathur**: manuscript editing.
**Shawn Roselle**: manuscript editing.





**Kenneth Pickering:** manuscript editing.
**Dale Allen:** manuscript editing.

**Acknowledgement**:
The authors thank Brian Eder, Golam Sarwar, and Janet Burke (U.S. /EPA) for their
constructive comments and suggestions during the internal review process.

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



Table 1. Statistics of DM8HR O$_3$ for all model cases over the domain and analysis regions in July 2011. The best performance metrics among the model cases are highlighted in bold.

| Region | Case | Record | OBS (ppb) | MOD (ppb) | RMSE (ppb) | NME (%) | MB (ppb) | NMB (%) | R |
|---|---|---|---|---|---|---|---|---|---|
| Domain | Base | 36242 | 48.21 | 52.04 | 12.6 | 19.2 | **3.8** | **8.0** | 0.69 |
| | mNLDN | 36242 | 48.21 | 53.40 | 12.9 | 19.8 | 5.2 | 10.8 | 0.70 |
| | hNLDN | 36242 | 48.21 | 52.21 | **11.9** | **18.4** | 4.0 | 8.3 | **0.72** |
| | pNLDN | 36242 | 48.21 | 52.52 | 12.7 | 19.5 | 4.3 | 8.9 | 0.70 |
| NE | Base | 5512 | 50.97 | 55.08 | 13.0 | 17.8 | 4.1 | 8.1 | 0.74 |
| | mNLDN | 5512 | 50.97 | 55.77 | 13.4 | 18.5 | 4.8 | 9.4 | 0.74 |
| | hNLDN | 5512 | 50.97 | 54.23 | **11.9** | **16.7** | **3.3** | **6.4** | **0.75** |
| | pNLDN | 5512 | 50.97 | 55.32 | 13.1 | 18.0 | 4.4 | 8.5 | 0.74 |
| SE | Base | 7061 | 44.55 | 51.71 | **12.6** | **21.0** | **7.2** | **16.1** | 0.76 |
| | mNLDN | 7061 | 44.55 | 53.33 | 13.6 | 236 | 8.8 | 19.7 | 0.76 |
| | hNLDN | 7061 | 44.55 | 52.30 | 12.6 | 21.7 | 7.8 | 17.4 | **0.77** |
| | pNLDN | 7061 | 44.55 | 52.39 | 13.0 | 22.0 | 7.8 | 17.6 | 0.76 |
| UM | Base | 8072 | 51.60 | 58.99 | 13.6 | 18.8 | 7.4 | 14.3 | 0.64 |
| | mNLDN | 8072 | 51.60 | 60.14 | 14.4 | 20.5 | 8.5 | 16.6 | 0.64 |
| | hNLDN | 8072 | 51.60 | 58.35 | **12.8** | **18.0** | **6.8** | **13.1** | 0.64 |
| | pNLDN | 8072 | 51.60 | 59.42 | 13.9 | 19.4 | 7.8 | 15.1 | 0.64 |
| LM | Base | 3609 | 42.15 | 46.21 | 12.4 | 21.5 | **4.1** | **9.6** | 0.73 |
| | mNLDN | 3609 | 42.15 | 47.93 | 12.9 | 22.3 | 5.8 | 13.7 | 0.74 |
| | hNLDN | 3609 | 42.15 | 47.12 | **12.3** | **21.3** | 5.0 | 11.8 | **0.76** |
| | pNLDN | 3609 | 42.15 | 46.93 | 12.6 | 21.8 | 4.8 | 11.3 | 0.74 |
| RM | Base | 6256 | 52.52 | 48.13 | 11.3 | 17.0 | -4.4 | -8.4 | 0.52 |
| | mNLDN | 6256 | 52.52 | 50.93 | 10.2 | 14.7 | **-1.6** | **-3.0** | 0.56 |
| | hNLDN | 6256 | 52.52 | 50.35 | **9.9** | **14.4** | -2.2 | -4.1 | **0.57** |
| | pNLDN | 6256 | 52.52 | 48.93 | 10.9 | 16.2 | -3.6 | -6.9 | 0.53 |
| PC | Base | 5570 | 44.72 | 47.58 | 11.7 | 20.1 | 2.9 | 6.4 | 0.80 |
| | mNLDN | 5570 | 44.72 | 47.73 | 11.6 | 20.0 | 3.0 | 6.7 | 0.80 |
| | hNLDN | 5570 | 44.72 | 46.65 | **11.3** | **19.5** | **1.9** | **4.3** | **0.81** |
| | pNLDN | 5570 | 44.72 | 47.62 | 11.6 | 20.0 | 2.9 | 6.5 | 0.80 |





Table 2. Statistics of daily mean NO$_x$ for all model cases over the domain and analysis regions July 2011. The best performance metrics among the model cases are highlighted in bold.

| Region | Case | Record | OBS (ppb) | MOD (ppb) | RMSE (ppb) | NME (%) | MB (ppb) | NMB (%) | R |
|--------|------|--------|-----------|-----------|------------|---------|----------|---------|-----|
| Domain | Base | 6912 | 7.58 | 8.88 | **8.7** | 62.6 | **1.3** | 17.1 | 0.54 |
| | mNLDN | 6912 | 7.58 | 8.87 | **8.7** | **62.5** | **1.3** | **17.1** | 0.54 |
| | hNLDN | 6912 | 7.58 | 8.92 | 8.7 | 62.7 | 1.3 | 17.7 | **0.55** |
| | pNLDN | 6912 | 7.58 | 8.87 | **8.7** | 62.5 | **1.3** | 17.1 | 0.54 |
| NE | Base | 989 | 10.48 | 9.72 | **7.0** | 46.0 | -0.8 | -7.3 | 0.55 |
| | mNLDN | 989 | 10.48 | 9.71 | **7.0** | **46.0** | -0.8 | -7.3 | **0.55** |
| | hNLDN | 989 | 10.48 | 9.77 | 7.1 | 46.1 | **-0.7** | **-6.8** | 0.55 |
| | pNLDN | 989 | 10.48 | 9.72 | **7.0** | 46.0 | -0.8 | -7.3 | 0.55 |
| SE | Base | 645 | 6.44 | 9.18 | 7.2 | 75.3 | 2.7 | 42.6 | 0.34 |
| | mNLDN | 645 | 6.44 | 9.17 | **7.2** | **75.1** | **2.7** | **42.4** | 0.34 |
| | hNLDN | 645 | 6.44 | 9.18 | 7.2 | 75.3 | 2.7 | 42.6 | **0.34** |
| | pNLDN | 645 | 6.44 | 9.17 | 7.2 | 75.2 | 2.7 | 42.5 | 0.34 |
| UM | Base | 542 | 11.42 | 18.09 | **18.7** | **82.7** | **6.7** | **58.4** | 0.58 |
| | mNLDN | 542 | 11.42 | 18.10 | 18.7 | 82.8 | 6.7 | 58.5 | **0.58** |
| | hNLDN | 542 | 11.42 | 18.22 | 18.9 | 83.6 | 6.8 | 59.5 | **0.58** |
| | pNLDN | 542 | 11.42 | 18.09 | 18.7 | **82.7** | **6.7** | 58.4 | **0.58** |
| LM | Base | 1240 | 6.11 | 8.32 | 6.0 | 61.2 | 2.2 | 36.1 | **0.68** |
| | mNLDN | 1240 | 6.11 | 8.30 | **6.0** | **61.1** | **2.2** | **35.9** | **0.68** |
| | hNLDN | 1240 | 6.11 | 8.33 | 6.0 | 61.3 | 2.2 | 36.3 | **0.68** |
| | pNLDN | 1240 | 6.11 | 8.31 | 6.0 | 61.2 | 2.2 | 36.0 | **0.68** |
| RM | Base | 1370 | 3.90 | 4.00 | 3.7 | 60.0 | **0.1** | **2.4** | 0.58 |
| | mNLDN | 1370 | 3.90 | 4.01 | **3.7** | **59.9** | **0.1** | 2.6 | 0.58 |
| | hNLDN | 1370 | 3.90 | 4.02 | 3.7 | 60.0 | 0.1 | 3.3 | **0.58** |
| | pNLDN | 1370 | 3.90 | 4.00 | 3.7 | 60.0 | **0.1** | **2.4** | 0.58 |
| PC | Base | 2056 | 8.61 | 9.52 | **9.1** | **62.8** | **0.9** | **10.6** | 0.48 |
| | mNLDN | 2056 | 8.61 | 9.52 | **9.1** | **62.8** | **0.9** | **10.6** | 0.48 |
| | hNLDN | 2056 | 8.61 | 9.59 | 9.1 | 62.9 | 1.0 | 11.4 | **0.48** |
| | pNLDN | 2056 | 8.61 | 9.52 | **9.1** | **62.8** | **0.9** | **10.6** | 0.48 |



Table 3. Statistics of June-August 2011 accumulated precipitation (cm) and wet deposition of nitrate ($NO_3^-$) for all model cases over the domain. The best performance metrics among the model cases are highlighted in bold.

| Region | Case | Record | OBS (cm, kg/ha) | MOD (cm, kg/ha) | RMSE (cm, kg/ha) | NME (%) | MB (cm, kg/ha) | NMB (%) | R |
|---|---|---|---|---|---|---|---|---|---|
| Domain | precip | 196 | 24.8 | 23.9 | 7.5 | 23 | -0.9 | -4 | 0.87 |
| | Base | 196 | 2.34 | 1.52 | 1.1 | 38 | -0.8 | -35 | 0.84 |
| | mNLDN | 196 | 2.34 | 1.98 | **0.8** | **26** | **-0.4** | **-15** | **0.86** |
| | hNLDN | 196 | 2.34 | 1.95 | **0.8** | **26** | **-0.4** | -17 | **0.86** |
| | pNLDN | 196 | 2.34 | 1.68 | 1.0 | 33 | -0.7 | -28 | 0.85 |
| NE | precip | 31 | 38.6 | 35.9 | 9.5 | 19 | -2.7 | -7 | 0.79 |
| | Base | 31 | 2.96 | 2.32 | 1.1 | 29 | -0.6 | -23 | 0.70 |
| | mNLDN | 31 | 2.96 | 2.71 | **0.9** | **24** | -0.3 | -8 | **0.76** |
| | hNLDN | 31 | 2.96 | 2.74 | **0.9** | **24** | **-0.2** | **-7** | 0.74 |
| | pNLDN | 31 | 2.96 | 2.48 | 1.0 | 27 | -0.5 | -16 | 0.73 |
| SE | precip | 39 | 36.1 | 31.7 | 9.4 | 21 | -4.3 | -12 | 0.80 |
| | Base | 39 | 3.05 | 2.09 | 1.2 | 35 | -1.0 | -32 | 0.51 |
| | mNLDN | 39 | 3.05 | 2.97 | **0.8** | **21** | **-0.1** | **-2** | **0.56** |
| | hNLDN | 39 | 3.05 | 2.82 | 0.9 | 23 | -0.2 | -8 | 0.53 |
| | pNLDN | 39 | 3.05 | 2.43 | 1.0 | 27 | -0.6 | -20 | 0.54 |
| UM | precip | 45 | 28.8 | 26.1 | 6.8 | 20 | -2.7 | -9 | 0.51 |
| | Base | 45 | 3.17 | 1.98 | 1.4 | 38 | -1.2 | -38 | 0.73 |
| | mNLDN | 45 | 3.17 | 2.51 | **0.9** | **24** | **-0.7** | **-21** | **0.77** |
| | hNLDN | 45 | 3.17 | 2.48 | **0.9** | 25 | **-0.7** | -22 | **0.77** |
| | pNLDN | 45 | 3.17 | 2.15 | 1.2 | 33 | -1.0 | -32 | 0.76 |
| LM | precip | 12 | 12.3 | 10.4 | 4.1 | 29 | -2.0 | -16 | 0.90 |
| | Base | 12 | 1.44 | 0.85 | 0.7 | 41 | -0.6 | -41 | 0.90 |
| | mNLDN | 12 | 1.44 | 1.16 | **0.6** | 33 | **-0.3** | **-19** | 0.88 |
| | hNLDN | 12 | 1.44 | 1.13 | **0.6** | **32** | **-0.3** | -21 | **0.89** |
| | pNLDN | 12 | 1.44 | 0.93 | 0.7 | 36 | -0.5 | -35 | 0.88 |
| RM | precip | 50 | 13.7 | 18.2 | 6.9 | 39 | 4.4 | 32 | 0.91 |
| | Base | 50 | 1.63 | 0.8 | 1.0 | 51 | -0.8 | -51 | 0.90 |
| | mNLDN | 50 | 1.63 | 1.1 | **0.7** | 34 | **-0.5** | -32 | **0.91** |
| | hNLDN | 50 | 1.63 | 1.12 | **0.7** | **33** | **-0.5** | **-31** | 0.90 |
| | pNLDN | 50 | 1.63 | 0.86 | 1.0 | 48 | -0.8 | -47 | **0.91** |
| PC | precip | 19 | 7.01 | 6.53 | **2.4** | 29 | **-0.48** | -6.8 | 0.84 |
| | Base | 19 | 0.31 | 0.31 | **0.18** | 44 | **0.00** | -1.0 | 0.88 |
| | mNLDN | 19 | 0.31 | 0.33 | 0.19 | 48 | 0.01 | 3.9 | **0.89** |
| | hNLDN | 19 | 0.31 | 0.33 | 0.20 | 50 | 0.02 | 6.6 | **0.89** |
| | pNLDN | 19 | 0.31 | 0.31 | **0.18** | 44 | **0.00** | **-0.3** | 0.88 |



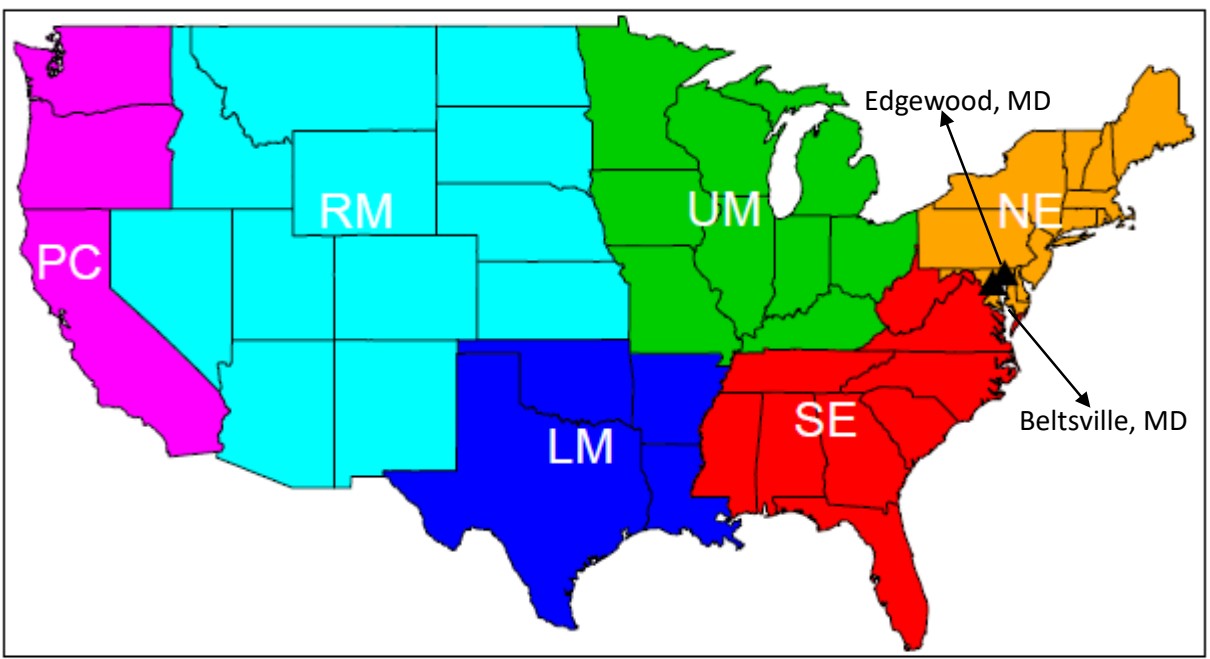

Figure 1. Analysis regions and ozonesonde locations during the 2011 DISCOVER-AQ field study.



Figure 2. Timeseries of regional-mean daily maximum 8-hr $O_3$ comparing observations (AQS) and CMAQ model predictions using the $LNO_x$ schemes to Base simulation for the domain (a), SE (b), and RM (c) in July, 2011. The numbers in the parentheses following the region names are the number of AQS sites.





Figure 3. Timeseries of daily mean NO$_X$ over the domain (a), SE (b), and RM (c) in July, 2011. The numbers in the parentheses following the region names are the number of AQS sites.



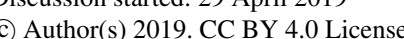


Figure 4. Diurnal profiles for hourly $O_3$ and $NO_X$ over the domain (a,d), SE (b,e), and RM (c,f) in July, 2011.



Figure 5. Spatial maps of the mean bias of DM8HR $O_3$ (model – observation) differences between model case with $LNO_x$ and the Base as well as the corresponding histograms indicating the number of sites with decreased mean bias for each pair of model cases in July, 2011.

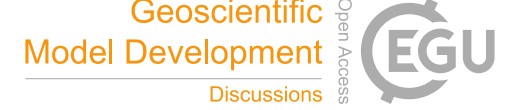

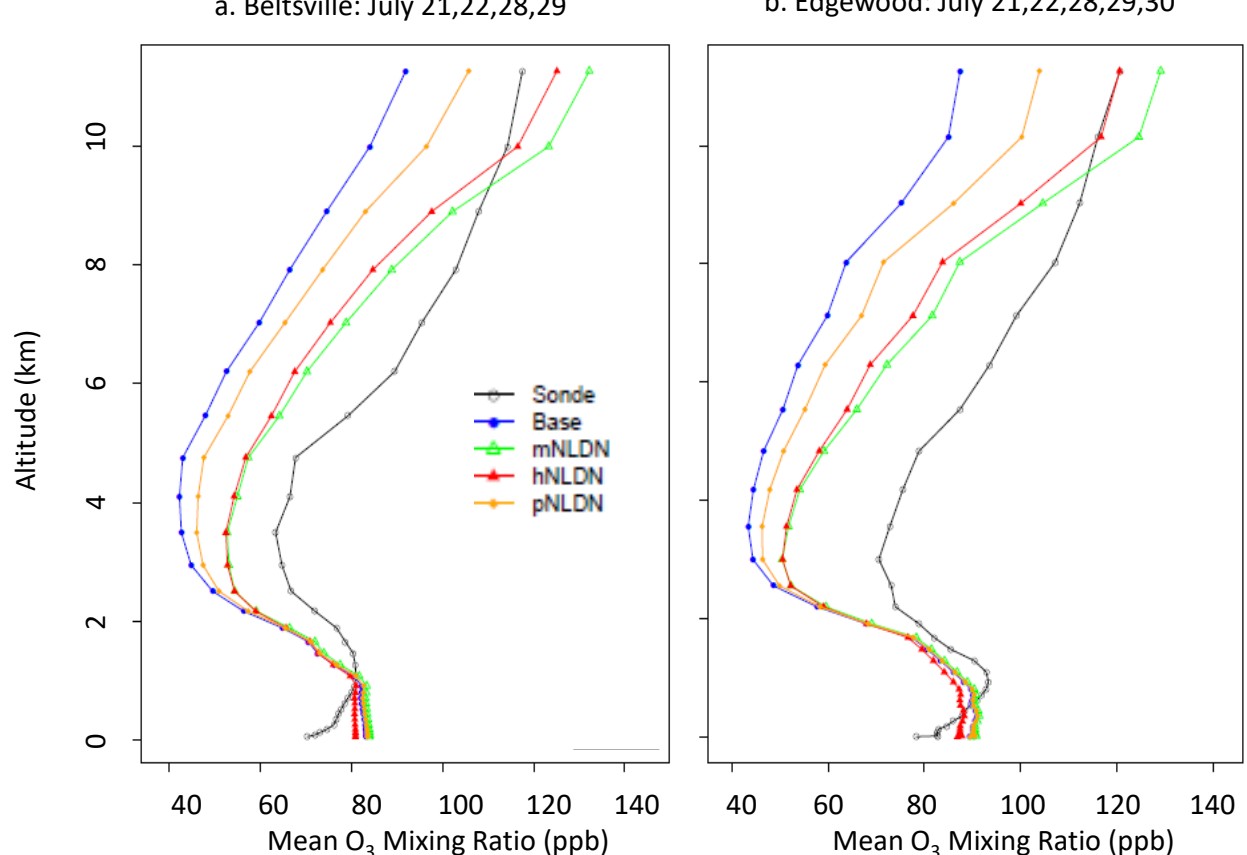

Figure 6. Vertical profiles of $O_3$ mixing ratios from ozonesonde measurements and model simulations at Beltsville, MD (a) and Edgewood, MD (b) on the days when lightning NO produced significant impact on $O_3$ during the Discover-AQ field study in July, 2011.





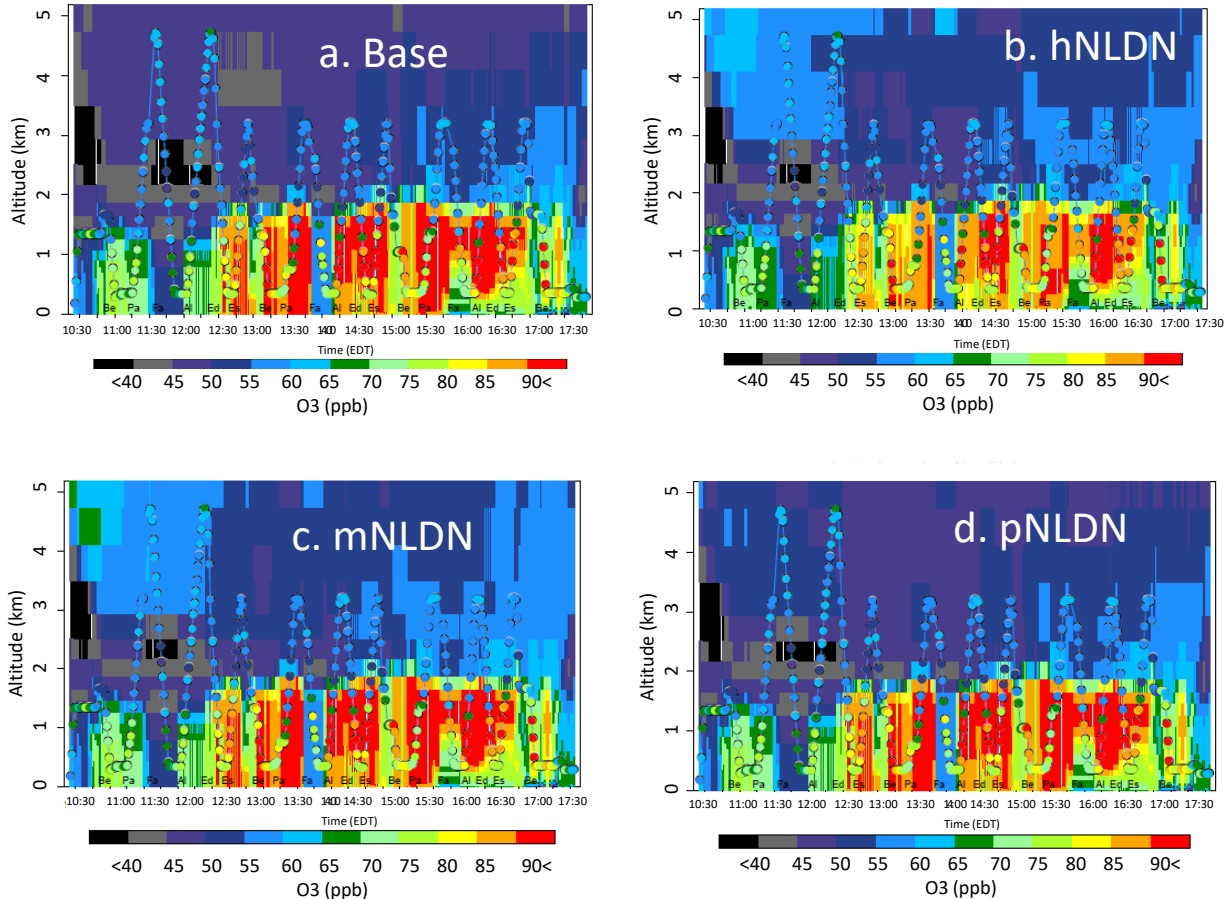

Figure 7. Overlay of P3B observed O$_3$ (1 minute mean values) over the corresponding vertical cross sections of simulated values extracted at the flying locations on July 28, 2018, (a) Base, (b) hNLDN,(c) mNLDN, and (d) pNLDN. The letters marked at the bottom of the plots are P3B spiral sites, Be: Beltsville, Pa: Padonia, Fa: Fairhill, Al: Aldino, Ed: Edgewood, Es: Essex.





Figure 8. The vertical-time difference between hNLDN and Base during the
P3B flight period on July 28, 2011 for (a) NO, (b) $NO_X$, and (c) $O_3$.





Figure 9. The vertical-time difference between mNLDN and Base during the P3B flight period on July 28, 2011 for (a) NO, (b) $NO_X$, and (c) $O_3$.



Figure 10. The vertical-time difference between pNLDN and Base during the P3B flight period on July 28, 2011 for (a) NO, (b) $NO_X$, and (c) $O_3$.





Figure 11. Bias (model – observation) distributions of O$_3$ (a) and NO (b) at each P3B spiral site on July 21, 22, 28, and 29, 2011. Be: Beltsville, Pa: Padonia, Fa: Fairhill, Al: Aldino, Ed: Edgewood, Es: Essex, Cb: Chesapeake Bay.





Figure 12. The top row shows precipitation estimates from WRF (left), the bias in the WRF predicted precipitation at NTN locations (middle) , and the corresponding scatter plots (right). The middle row shows wet deposition (Dep) of nitrate estimates from the Base simulation (left), the bias in the Base model estimates of wet deposition of $NO_3^-$ at NADP/NTN locations (middle), and the corresponding scatter plots (right). The bottom row shows the difference in the $LNO_x$ sensitivity simulations and the Base case estimates of wet deposition of $NO_3^-$: mNLDN – Base (left); hNLDN – Base (middle), and pNLDN – Base (right) . All maps are based on accumulated values (precipitation or wet deposition) during June – August 2011. Precipitation totals are in cm and wet deposition totals are in kg/ha.