# Peer review of "Simulating Lightning NO Production in CMAQv5.2: Performance Evaluations Daiwen Kang1\*, Kristen M. Foley1, Rohit Mathur1, Shawn J. Roselle1, Kenneth E. Pickering2, and Dale J. Allen2 1Center for Environmental Measurement & Modeling, U.S. Environmental Protection Agency, Research Triangle Park, NC 27711, USA 2Department of Atmospheric and Oceanic Science, University of Maryland, College Park, MD, U"

_Geoscientific Model Development, 2019_

## Referee Comment (RC1) · Anonymous Referee #1 · 1 Jul 2019

This article appears to be the evaluation side of GMD-2019-33 "Simulating Lightning NOX Production in CMAQv5.2: 2 Evolution of Scientific Updates", which is cited numerous times in the manuscript, and with similar authors (although the order is not exactly the same). I would suggest to make the link more specific and make these two papers companion papers, possibly entitled "Simulating Lightning NOX Production in CMAQv5.2: part 1, new parameterizations", and part 2: evaluation for example.

The paper is well written, concise, and of good scientific quality, with a thorough evaluation of the impact of the three new schemes that have been implemented into CMAQ. I have a few remarks that should in my opinion be addressed before final publication:

[Figure]

• Please add a short descriptive summary of the three lightning schemes that are evaluated in the paper,

• It would be desirable to remind the reader of the different chemical links between NOx, O3 and nitrate precursors; this is partially done at the beginning of section 3.3 for nitrate.

• Perhaps a discussion on the skill of the forecasts of convective precipitations in the WRF forecast (and possibly of its diurnal cycle) should be discussed or at least mentioned since this is a critical input of the three schemes,

• For nitrate, perhaps it would have been simpler to evaluate the nitrate concentrations against observations from the CASTNET network, rather than nitrate wet deposition, which depends again on modelled precipitation: this adds another layer of error/uncertainty.

• Tables 1 and 2 are very big; the bold parts are not always easy to spot. Is there a way to present this key information in graphics?
* * *

---

## Short Comment (SC1) · 15 Jul 2019

Thank you for the efforts you have gone to to provide transparent access to the code and data for this manuscript. There are, however, a few points at which the code and data availability section is not currently compliant with GMD policy. These need to be remedied in the revised manuscript.

1. The CMAQ and WRF code references point to project websites. This is insufficiently persistent and precise for GMD purposes. Please also cite a persistent public archive of the exact version of the source used.

[Figure]

2. Data processing and analysis scripts are available "on request". This does not meet GMD requirements. Please provide a citation of a persistent public archive of the scripts (e.g. Zenodo).

3. The lightning dataset is proprietary, which is acceptable. However please identify exactly the data set and version used so that a reader who wished to reproduce the work would know exactly what they needed to purchase and use.

4. The data citation to Zenodo is excellent. Please ensure that the additional data which is only "on request" is not actually required to reproduce the results.

Further details of the requirements for code availability can be found at https://www. geoscientific-model-development.net/about/code_and_data_policy.html. That page has been updated since this manuscript was submitted, however with respect to the issues in this manuscript, the new version is merely a more clearly stated version of what is required, rather than a substantive change in policy.

---

## Referee Comment (RC2) · Anonymous Referee #2 · 31 Jul 2019

General Comments

The scope of this paper is to evaluate the impact of three Lightning NOx parameterisation schemes in WRF-CMAQ on ozone, NOx and nitrate deposition compared with a base case without such parameterisation. The use of a variety of observations at different heights is commendable and it is clearly presented. The paper is well written and easy to follow. Although differences between the three parameterisation schemes and the base case are generally large, the three schemes perform fairly similarly to each other in a number of cases presented. This is not surprising, given that the three parameterisation schemes used are different versions of the same scheme. However,

the authors use all the observations in their toolbox to provide a clear explanation of where the schemes show the largest difference and try to identify the best performing scheme.

Specific Comments

There is not enough information about the three parameterisation schemes. It would be useful to add at least a very short description here (including the vertical distribution algorithm) and then refer the reader to the relevant paper for further details.

Given that the model uses hourly or monthly observed lightning flashes information from the NLDN network, I expect this parameterisation schemes are only available for simulations of the past, e.g. hindcasts and case studies, but not for air quality forecasts (for which the observed lightning flashes are not available). Can the authors add a comments in the text to address the relevance of this work for air quality forecast or specify its intended areas of application?

l.184-185 "...all model cases with LNOx exhibit slightly higher correlation coefficients than the base simulation, suggesting the importance..." Looking at table 1 and 2 I see identical values for most locations and tiny differences (0.69 vs 0.70; 0.73 vs 0.74; 0.52 vs 0.53) for other cases. I would rather say that the correlation coefficients between simulations with and without LNOx are not significantly different!

l.257-259 can the authors comment on why NOx is over/under-estimated during night/day-time?

In Figure 4, the legend for AQS is wrong (no star symbol used in the plots)

Figure 6. It would be interesting to add 2 further panels to show equivalent results for NOx profiles in the different model simulations. Can this help explain the lower surface ozone in hNLDN? If not, can the author suggest what processes are responsible for it?

Technical Comments

l.53 "pNLDN, provides an improved estimate for LNOx compared to the base simulation that does not include LNOx." LNOx is of course improved if it is included in the simulation! I think this should be: "...provides an improvement for ozone and NOx compared to the base simulation..."

l.65-66 "The significant impact of LNOx on surface air quality was earlier..." Given the explanation given by the authors I think this should be: "The significant impact of LNOx on process-based understanding of surface air quality was earlier..."

l.66 replace "in that" with "which found"

l.288-289 "the vertical profile lines can be separated" this is confusing, replace with same text used later (l.308) which is much clearer.

---

## Author Comment (AC1) · 6 Aug 2019

Thank you for pointing out the loose points for the code and data accessibility. We have now provided clarity in the manuscript with more accurate links and descriptions.

1. The CMAQ and WRF code references point to project websites. This is insufficiently persistent and precise for GMD purposes. Please also cite a persistent public archive of the exact version of the source used.

We have now updated the links for both WRF and CMAQ to point to their specific versions.

[Figure]

WRF: http://www2.mmm.ucar.edu/wrf/users/wrfv3.8/updates-3.8.html

CMAQ: https://github.com/USEPA/CMAQ/tree/5.2

2. Data processing and analysis scripts are available "on request". This does not meet GMD requirements. Please provide a citation of a persistent public archive of the scripts (e.g. Zenodo).

We have now updated the dataset with all the scripts used to create the tables and plots in the manuscript as the dataset version 2.0: https://zenodo.org/record/3360744

3. The lightning dataset is proprietary, which is acceptable. However please identify exactly the data set and version used so that a reader who wished to reproduce the work would know exactly what they needed to purchase and use.

Unfortunately, we don't have additional information about the data set and its version. To obtain this dataset, one needs to contact Vaisala Inc. directly, and they would prepare the data with the region and time period from their database and that data is the original lightning flash data collected and managed by Vaisala Inc. For clarity, we added the sentence "The lightning data obtained from Vaisala Inc. is the cloud-to-ground lightning flashes over the contiguous United States".

4. The data citation to Zenodo is excellent. Please ensure that the additional data which is only "on request" is not actually required to reproduce the results.

In the data citation to Zenodo, we have provided the immediate data tables to produce the tables and plots without the "on request" data. However, the scripts provided could directly use the immediate data tables or start from the original "on request" data to understand how the immediate data tables are generated.

---

## Author Comment (AC2) · 13 Aug 2019

This article appears to be the evaluation side of GMD-2019-33 "Simulating Lightning NOX Production in CMAQv5.2: 2 Evolution of Scientific Updates", which is cited numerous times in the manuscript, and with similar authors (although the order is not exactly the same). I would suggest to make the link more specific and make these two papers companion papers, possibly entitled "Simulating Lightning NOX Production in CMAQv5.2: part 1, new parameterizations", and part 2: evaluation for example.

**We agree with the reviewer, and we have now changed the title to "Simulating Lightning NO Production in CMAQv5.2: Performance Evaluations" to match the companion paper that has been published.**

The paper is well written, concise, and of good scientific quality, with a thorough evaluation of the impact of the three new schemes that have been implemented into CMAQ.

**We thank the reviewer for the overall positive assessment.**

I have a few remarks that should in my opinion be addressed before final publication:

âA˘ c Please add a short descriptive summary of the three lightning schemes that are ´ evaluated in the paper

**The other reviewer also made the same suggestion, so we have now added this information in the Methodology section as "2.1 The LNO schemes" in the revised manuscript on Page 4.**

âA˘ c It would be desirable to remind the reader of the different chemical links between ´ NOx, O3 and nitrate precursors; this is partially done at the beginning of section 3.3 for nitrate.

**We thank the reviewer for this suggestion. Even though the role of NOx in the atmospheric chemistry is well known, we agree that for completeness it would be useful to briefly summarize the role for broader readership. We have now incorporated the information in the introduction section by stating that "it is expected that the relative contribution of LNO to the tropospheric $NO_x$ burden and its subsequent impacts on atmospheric chemistry as one of the key precursors for ozone ($O_3$), hydroxyl radical (OH), nitrate, and other species will increase in the United States and other developed countries".**

âA˘ c Perhaps a discussion on the skill of the forecasts of convective precipitations in ´ the WRF forecast (and possibly of its diurnal cycle) should be discussed or at least mentioned since this is a critical input of the three schemes,

**The reviewer makes a good point. However, there are no observations to distinguish convective precipitation from non-convective precipitation, and usually for precipitation measurements, only aggregated daily or even longer-period products such as PRISM (a combination of rain gauge and modeling results) and STAGE (a combination of radar and rain gauge observations) are available. Therefore, it is not readily possible to assess the forecast skills for convective precipitation, and even more so for the diurnal cycle. For this**

**reason, we provided the monthly accumulated precipitation assessment in Figure 12 for WRF precipitation and computed the statistics over the NTN sites to form the basis for our nitrate wet deposition evaluation.**

âAˇ c For nitrate, perhaps it would have been simpler to evaluate the nitrate concentrations against observations from the CASTNET network, rather than nitrate wet deposition, which depends again on modelled precipitation: this adds another layer of error/uncertainty.

**We agree that the CASTNET network offers another source of evaluation for model estimated nitrate. The advantage of using the NADP/NTN network is that it is larger with more spatial coverage (196 NTN sites compared to about 75 CASTNET sites). The top row in Figure 12 provides the model bias in annual total precipitation compared to NTN observations so that it can be compared against the wet deposition bias. These plots indicate that, while there is some underestimation of precipitation in the eastern half of the US, errors in modeled precipitation do not account for 35% normalized mean bias in modeled wet deposition of nitrate across the country. This consistent bias suggests missing regional-scale emissions sources such as NO from lightning. Additionally, direct evaluation of modeled wet deposition estimates is used to inform national scale assessments of nitrogen budgets and comparison of deposition loads with critical loads. The evaluation again the NTN network is used to demonstrate how the addition of NO from lightning can help reduce bias in modeled nitrate deposition levels, increasing the credibility of using model output for critical loads analyses. It should also be noted that comparisons of aerosol NO3 predictions with ambient observations (from CASTNET and other networks) can be influenced by errors in modeled gas/aerosol partitioning influenced by uncertainties in NH3 emissions. Comparisons of total NO3 wet deposition also helps circumvent those other model error influences and better isolate the impacts of LNO emissions on total nitrate atmospheric budget.**

âAˇ c Tables 1 and 2 are very big; the bold parts are not always easy to spot. Is there a ´ way to present this key information in graphics?

**We agree that the tables contain a lot of information, but as a supplementary to the information that has presented in other graphics and plots, interested readers can get more detailed information from the tables for model performance over different geographical regions.**

---

## Author Comment (AC3) · 13 Aug 2019

General Comments

The scope of this paper is to evaluate the impact of three Lightning NOx parameterisation schemes in WRF-CMAQ on ozone, NOx and nitrate deposition compared with a base case without such parameterisation. The use of a variety of observations at different heights is commendable and it is clearly presented. The paper is well written and easy to follow. Although differences between the three parameterisation schemes and the base case are generally large, the three schemes perform fairly similarly to each other in a number of cases presented. This is not surprising, given that the three parameterisation schemes used are different versions of the same scheme. However, the authors use all the observations in their toolbox to provide a clear explanation of where the schemes show the largest difference and try to identify the best performing scheme.

**We thank the reviewer for the overall positive assessment of the manuscript.**

 Specific Comments

There is not enough information about the three parameterisation schemes. It would be useful to add at least a very short description here (including the vertical distribution algorithm) and then refer the reader to the relevant paper for further details.

**Both reviewers have suggested including additional details on the LNO schemes. In the revised manuscript, we have now added this information to the Methodology Section as "2.1 The LNO schemes" on Page 4 of the revised manuscript.**

 Given that the model uses hourly or monthly observed lightning flashes information from the NLDN network, I expect this parameterisation schemes are only available for simulations of the past, e.g. hindcasts and case studies, but not for air quality forecasts (for which the observed lightning flashes are not available). Can the authors add a comments in the text to address the relevance of this work for air quality forecast or specify its intended areas of application?

**In this study, three lightning NO schemes are involved. It is correct, all the schemes are related to the observed lightning flashes from NLDN network, but the formulations are different. The hourly (hNLDN) or monthly (mNLDN) schemes do depend on the availability of the observed NLDN data for their applications, but the third one, the parameterized scheme (pNLDN), was derived using historical data from the observed NLDN data and model predicated convective precipitation, and its application doesn't require the actual observed data. Instead, the lightning flashes are derived from the linear and log-linear relationship that is parameterized in the scheme. And it is specifically tailored for applications such as air quality forecast when the observed lightning flashes are not available. We have now incorporated this point in Conclusions on Page 15 of this revised manuscript.**

l.184-185 "...all model cases with LNOx exhibit slightly higher correlation coefficients than the base simulation, suggesting the importance..." Looking at table 1 and 2 I see identical values for most locations and tiny differences (0.69 vs 0.70; 0.73 vs 0.74; 0.52 vs 0.53) for other cases. I

would rather say that the correlation coefficients between simulations with and without LNOx are not significantly different!

**Though the difference between correlation coefficients are small, but the increase is persistent through the domain and all subregions that indicates the general trend. Therefore, we describe it as slightly higher.**

l.257-259 can the authors comment on why NOx is over/under-estimated during night/day-time?

**The question of why NOx is over/under-estimated during night/day-time is rather a complicated issue that is currently under active investigation in many research groups with coordinated efforts. There are several hypotheses including (1) issues related to representation of vertical mixing, (2) issues related to magnitude of anthropogenic emissions, especially from the mobile sector, and (3) spatial and temporal allocation of emissions.**

In Figure 4, the legend for AQS is wrong (no star symbol used in the plots)

**We thank the reviewer for catching this error. It has now been corrected.**

Figure 6. It would be interesting to add 2 further panels to show equivalent results for NOx profiles in the different model simulations. Can this help explain the lower surface ozone in hNLDN? If not, can the author suggest what processes are responsible for it?

**We thank the reviewer for the suggestion. The impact of lightning NO$_x$ on O$_3$ production generally occurs downwind of the location of lightning flashes as revealed in our later analysis related to Figures 7-10. Often it is the case that when the ozonesonde measurements indicated difference on O$_3$ mixing ratios, the difference of NO$_x$ mixing ratios from the different model cases is insignificant at the same location. We however examine the issue raised by the reviewer using the aircraft measurements in detail in later section in the manuscript.**

Technical Comments

l.53 "pNLDN, provides an improved estimate for LNOx compared to the base simulation that does not include LNOx." LNOx is of course improved if it is included in the simulation! I think this should be: "...provides an improvement for ozone and NOx compared to the base simulation..."

**Thanks. It was a typo. It has now been revised to "an improved estimate of nitrate wet deposition"**

l.65-66 "The significant impact of LNOx on surface air quality was earlier..." Given the explanation given by the authors I think this should be: "The significant impact of LNOx on process-based understanding of surface air quality was earlier..."

**Thanks. The sentence has been modified as suggested by the reviewer.**

l.66 replace "in that" with "which found"

**Thanks, the change has been made.**

l.288-289 "the vertical profile lines can be separated" this is confusing, replace with same text used later (l.308) which is much clearer.

**Thanks, we have revised the manuscript as suggested.**

---

## Author Response (AR2)

Comments to the Author:

Minor comment: I do not think that the surface statistics presented in Table 1 and 2 suggest "the importance of including the contributions of this source for improving the spatial and temporal variability in model predictions". They show a very marginal gain. It is not surprising because the base model is "tuned" (input parameters and schemes adjusted) to match as well as possible existing observations. Therefore, even by making the model more realistic in terms of emissions or physics, one may degrade initially the model performances. Overall, regarding the importance of LNO, results are much more convincing for the free troposphere. I do not think you need to add "suggesting…predictions" here. The point is made more naturally in the conclusion.

**The topical editor has made a very good point and we completely agree. We have now revised the manuscript by removing the sentence "suggesting … predictions" on Page 7 Lines 200 – 201. And made minor changes to make the sentence flow smoothly.**

[revised manuscript text omitted]